# Chlorination Treatment of Meta-Aramid Fibrids and Its Effects on Mechanical Properties of Polytetramethylene Ether Glycol/Toluene Diisocyanate (PTMEG/TDI)-Based Polyurethane Composites

**DOI:** 10.3390/polym11111794

**Published:** 2019-11-01

**Authors:** Wuyang Lu, Yuhua Yi, Chunping Ning, Mingliang Ge, Jahangir Alam S.M.

**Affiliations:** Department of Industrial Equipment and Control Engineering, School of Mechanical & Automotive Engineering, South China University of Technology, Guangzhou 510640, China; luwuyang1682@163.com (W.L.); ningchp@163.com (C.N.); gml@scut.edu.cn (M.G.)

**Keywords:** chlorination, meta-aramid fibrids, PTMEG/TDI, polyurethane composites

## Abstract

Meta-aramid fibrids (MAF) have attracted much attention. However, it is difficult for this high mechanical performance fiber to form sufficient interface adhesion between the MAF and polyurethane (PU) matrix due to the chemical inertness of its surface. Thus, the surface activity of MAF should be improved to obtain a high-performance MAF/PU composite. A novel methodology to modify the surface of MAF with a sodium dichloroisocyanurate solution (DCCNa) was developed to obtain chlorinated MAF (MAFC) in this study. A series of MAFC/PU composites was prepared by in situ polymerization processes. The results of Fourier-transform infrared spectroscopy (FTIR) and X-ray photoelectron spectroscopy (XPS) demonstrated that the chlorine-contained chemical groups were introduced onto the MAF surfaces after chlorination. Dynamic contact angle analysis (DCAA) revealed that the surface wettability and the surface free energy of the MAFC were significantly improved, which allowed for strong chemical bonding to PU. Scanning electron microscopy (SEM) showed a uniform distribution of MAFC and good interfacing bonding between the MAFC and PU. With the incorporation of 1.5 wt% MAFC into the polyurethane matrix, the tensile and tear strength values of MAFC/PU were 36.4 MPa and 80.1 kN·m^−1^ respectively, corresponding to improvements of approximately 43.3% and 21.1%, as compared to those of virgin PU as 25.4 MPa and 66.1 kN·m^−1^, respectively.

## 1. Introduction

It has been reported that polytetramethylene ether glycol/toluene diisocyanate (PTMEG/TDI)-based polyurethane (PU) elastomer is a block copolymer composed of alternating soft and hard segment phase, wherein the hard segment phase is composed of toluene diisocyanate (TDI) and a chain extender, and the soft segment phase is composed of polytetramethylene ether glycol (PTMEG) [1]. In the molecular structure of PTEMG, four methylene groups are regularly arranged between ether bonds and endow to form a semi-crystalline polymer [2,3]. Therefore, polyurethane elastomers prepared with PTMEG are widely used in dynamically demanding applications such as rolls, tires, and wheels because of their relatively high resistance to hydrolytic cleavage, high resilience, high wear resistance, good low-temperature flexibilities, good processabilities, and superior dynamic performance [4]. However, PTMEG/TDI-based PU elastomer cannot be used in high-load applications due to the insufficient mechanical strength of the PTMG2000/TDI polyurethane elastomer, here PTMEG2000 refers to the PTMEG with the average molecular weight of 2000. Thus, the mechanical properties of PTMEG/TDI-based PU elastomers should be improved.

In recent years, a variety of appropriate fiber fillers had been incorporated into polyurethane significantly improved the mechanical properties, including jute fibers [5], natural cellulose fibers [6], hemp fibers [7], carbon fibers [8], fiberglass [9], and aramid fibers [10]. The other researchers had defined that the meta-aramid fibrids (MAF) are a kind of aramid fiber prepared through precipitation by injecting the meta-aramid solution into a high-shear coagulation bath [11,12]. The MAF is a ribbon-like material with an average length of 0.2–1 mm, an aspect ratio ranging from 1000 to 3000, and a thickness less than 2 µm. Thus, MAF possesses large specific surface areas [13,14]. MAF also possesses the inherent properties of meta-aramid fibers, such as higher strength, higher modulation, and excellent thermal and chemical stabilities. As a result, MAF is widely used in high-temperature materials and honeycomb-structure materials.

Although MAF possesses many excellent properties, due to the surface chemical inertness and the steric effect of the benzene rings in the molecular chains, the amide groups of MAF do not react readily with other atoms or groups, which results in poor interface adhesion between the MAF and polyurethane matrix [15,16]. Therefore, it is vital to modify the surfaces of MAF to improve the interface adhesion with the matrix. The chemical modification is a relatively common surface treatment method for aramid fibers, which can improve the surface polarity of aramid fibers by introducing reactive groups and elements onto the surfaces of aramid fibers [17]. According to the literature, the chemical modification methods based on strong acid modification include phosphoric acid modification [15,17,18], fluorination modification [19,20], chlorination modification [21], and so on.

In this study, we used sodium dichloroisocyanurate (DCCNa) as a high-efficiency chlorination treatment agent for MAF to obtain chlorinated MAFC. MAF/PU and MAFC/PU composites were prepared by in situ polymerizations. The MAF and MAFC were characterized by Fourier-transform infrared spectroscopy (FTIR), X-ray photoelectron spectroscopy (XPS), dynamic contact angle measurements and scanning electron microscopy (SEM). The mechanical properties of the MAF/PU and MAFC/PU composites were characterized using tensile testing.

## 2. Materials and Methods

### 2.1. Materials

The meta-aramid fibrids (MAF) were supplied by Guangdong Feibo New Material Technology Co. Ltd., Yunfu, Guangdong, China; the 2,4-toluene diisocyanate (TDI-100) was purchased from Mitsui Chemicals, Inc, Tokyo, Japan; the polytetramethylene ether glycol (PTMEG, *M*n = 2000) was provided by the Mitsubishi Chemical Corporation, Kyoto, Japan; the 3,3′-dichloro-4,4′-diaminodiphenylmethane (MOCA) was obtained from the Suzhou Xiangyuan Special Fine Chemical Co. Ltd., Suzhou, Jiangsu, China. The raw materials were all industrial grade. Analytical grade ethyl acetate (EA) was supplied by the Shanghai Ourchem Biotechnology Co., Ltd., Shanghai, China. Sodium dichloroisocyanurate (DCCNa) was provided by the Jinan Delan Chemical Co., Ltd., Jinan, Shandong, China.

### 2.2. Chlorination Treatment of MAF

The surfaces of MAF were cleaned with an ethyl acetate (EA) solution. To achieve a better cleaning effect, the mixed solution of MAF and EA were dispersed at 4000 rpm for 15 min using a high-speed disperser. The MAF was subsequently extracted by a vacuum filtration device. The MAF was cleaned two to three times using this method, and the MAF were fully dried in an oven at 110 °C for 4 h. The dried MAF was dispersed by a high-shear force using a 2800-rpm high-speed pulverizer. The dispersed MAF was added to a 10% DCCNa solution, and after manual stirring for 5 min, the DCCNa solution was suction filtered. The MAF was cleaned with distilled water several times until the surface pH value of the MAF was neutral. The modified MAF (MAFC) was dried in the oven at 110 °C for 12 h.

### 2.3. Preparation of MAFC/PU Composites

MAFC/PU composites were prepared by in situ polymerizations. First, the pretreated MAFC was added to the PTMEG2000 and dispersed at 4000 rpm for 15 min using a high-shear mixture. The mixture was added to a three-necked flask equipped with a stirring system, a vacuum system, and a thermometer. Vacuum dehydration of the mixture was performed at 110 °C for 2–3 h. When the moisture content of the mixture was less than 0.05%, as detected by a Karl Fischer Moisture Analyzer, the calculated TDI was added into the mixture and reacted at (80 ± 2) °C for 2 h to obtain a prepolymer. The NCO group content of the prepolymer was measured. Finally, the MOCA (which is called a chain extender, melted at 115 ± 3 °C) was added to the prepolymer, which was preheated to 80 ± 2 °C in advance; then, the mixture was rapidly stirred for 1–2 min, and poured into molds that were preheated to 110 °C in advance. After vulcanizing on a plate vulcanizing machine for 1 h, the MAFC/PU composite samples were taken out and placed in an electric blast drying oven at 100 °C for 20 h. The formulations of MAF/PU and MAFC/PU composites with different ratios of filler have been summarized in Table 1.

### 2.4. Characterization

SEM was conducted to investigate the morphologies of the MAF and MAFC using a Quanta FEG 250 scanning electron microscope (FEI, Hillsboro, OR, USA) operating under secondary electron mode at an accelerating voltage of 5 kV. The FTIR spectra of the MAF and MAFC were recorded on a Vertex-70 (Bruker, Karlsruhe, Germany) spectrometer with an attenuated total reflectance (ATR) in the scanning range of 500–4000 cm^−1^. XPS was performed to analyze the surface element change of the MAF and MAFC using an Axis Ultra DLD X-ray photoelectron spectrometer (Kratos Analytical, Manchester, UK) with a monochromatic Al Kα source (1486.6 eV).

The surface free energy and contact angles of the MAF and MAFC were evaluated using an OCA40 micro-dynamic contact angle analysis system (Data Physics Instruments, GmbH, Filderstadt, Germany). The surface free energy of the MAF and MAFC were measured using deionized water ($\gamma_{L}^{d}$ = 21.8 mN/m, $\gamma_{L}^{p}$ = 51 mN/m, and $\gamma_{L}$ = 72.8 mN/m) and ethylene glycol ($\gamma_{L}^{d}$ = 29.3 mN/m, $\gamma_{L}^{p}$ = 19 mN/m, and $\gamma_{L}$ = 48.3 mN/m) as testing liquids. The surface free energy can be calculated from the following Equations [22,23]:(1)$$\gamma_{L}^{}(1+\cos\theta )=2\sqrt{\gamma_{L}^{d}\gamma_{S}^{d}}+2\sqrt{\gamma_{L}^{p}\gamma_{S}^{p}}$$
(2)$$\gamma_{S}=\gamma_{S}^{d}+\gamma_{S}^{p}$$
where *θ* is the contact angle at the solid/liquid interface, and $\gamma_{L}$ and *γ_S_* are the surface tension of the testing liquid and fibrids, respectively. $\gamma_{L}^{d}$ and $\gamma_{L}^{p}$ are the polar and dispersive components of the surface tension of the testing liquid, respectively. $\gamma_{S}^{d}$ and $\gamma_{S}^{p}$ represent the polar component and the dispersive component of the total surface free energy of the fibrids, respectively [22,23].

Tensile tests of the composite samples were measured using an LD-24.104 electromechanical universal testing machine (LABSANS, Shenzhen, China). The measurements of the tensile strength and elongation at break were based on the ASTM D412 standard, and the rate of the extension was 500 mm/min. The tear strength was measured according to ASTM D624. Measurements of the hardness values of the composite samples were performed using an LX-A Model Hardness Meter (Shanghai Precision Instruments Co. Ltd., Shanghai, China) according to the ASTM D2240 standard; the rebound rate was measured using a WT B-0.5 impact testing machine according to the ASTM D2632 standard. The morphologies of the polyurethane composites were investigated using a Quanta FEG 250 SEM (FEI, Hillsboro, OR, USA).

## 3. Results and Discussion

### 3.1. Surface Morphologies of MAF and MAFC 

Figure 1 shows the SEM images of original fibrids (MAF) and modified fibrids (MAFC). As shown in Figure 1a,b, the original MAF was irregular ribbon-like or film-like materials due to the high-shear coagulation bath. The MAF had an average length of 0.2–1 mm with aspect ratios of 5:1–10:1. The thickness was less than 1–2 µm. Therefore, the MAF possessed large specific surface areas of soft shape and excellent flexibility but excellent toughening and mechanical properties. However, Figure 1a,b exhibits a rather smooth surface of MAF. By contrast, Figure 1c,d shows that the surface of MAFC became slightly rough, implying that the surfaces of MAF were successfully activated by the chlorination treatment.

### 3.2. IR Analysis

Figure 2 shows the FTIR spectra of MAF, DCCNa, and MAFC. The molecular structures of MAF, DCCNa, and MAFC were characterized by FTIR analysis. As shown in Figure 2, the characteristic absorption peaks of the MAF at 3415 cm^-1^ and 3060 cm^-1^ have corresponded to the N–H and aromatic C–H stretching vibrations, respectively, the vibration peaks of 1660 cm^−1^, 1608 cm^−1^, and 1540 cm^−1^ were attributed to C=O stretching vibrations, aromatic C=C stretching vibrations, and N–H deformation vibrations in MAF, respectively [11,24]. The characteristic peaks of DCCNa at 1742 cm^−1^, 1355 cm^−1^, and 700 cm^−1^ were assigned to C=O, C–N and N–Cl stretching vibrations, respectively. The new absorption peaks at approximately 3215 and 1753 cm^−1^ in the MAFC have corresponded to the –NH_2_ symmetrical stretching vibrations, and C=O stretching vibrations of –COOH, respectively, which indicated that hydrolysis reactions occurred on the surfaces of the MAFC during the chlorination modification. Furthermore, the new vibration peaks at approximately 530 cm^−1^ and 750 cm^−1^ for the MAFC corresponded to N–Cl stretching vibrations due to N-chlorination of the amidic nitrogen after the chlorination treatment. Lastly, the new peak at 1052 cm^−1^ was assigned to C–Cl vibrations, which was due to the Orton rearrangement of partially chlorinated N [25,26]. The entire chlorination treatment mechanism is illustrated in Figure 3.

### 3.3. Surface Chemical Composition of MAF and MAFC

Figure 4 shows the XPS wide-scan spectra and Cl 2p spectra of the MAF and MAFC. To demonstrate the mechanism of the chlorination treatment of the fibrids, XPS was used to investigate the chemical surface composition of MAF and MAFC. As shown in Figure 4 and Table 2, the MAF and MAFC contained inherent oxygen, nitrogen, and carbon elements. In addition, the presence of chlorine in the MAFC indicated that the MAF was chlorinated. The surface elemental analysis results for the MAF and MAFC are listed in Table 2. The chlorine contents of the MAF and MAFC were found 0 and 7.9%, respectively. It was also demonstrated that the chlorine was attached to the surfaces of MAF after chlorination modification. Compared with the MAF, it apparently decreased from 80.9% to 70.2% of carbon (C) concentration; and oxygen (O) concentration was increased from 9.6% to 12.7.7% with the increase in the oxygen/carbon (O/C) atomic ratio from 0.119 to 0.181; Furthermore, the nitrogen (N) concentration decreased slightly from 9.5% to 9.2% when the nitrogen/carbon (N/C) atomic ratio increased from 0.118 to 0.131. The increase in the N/C and O/C atomic ratios indicated that new polar chemical groups (–COOH and –NH_2_) were introduced to the MAFC surfaces by the chlorination treatment. To ascertain the state of chlorine (Cl) in the MAFC molecular structure after chlorine (Cl) treatment, the peaks of Cl (2p) were further investigated in Figure 4d. The Cl (2p) spectrum of the MAFC was curve fit with four peaks at binding energies of approximately 200.15 eV, 200.74 eV, 201.45 eV, and 202.35 eV [27]. The fit peaks at 200.15 eV and 200.74 eV were attributed to the Cl (2p^1/2^), and Cl(2p^3/2^) of O=C–N–Cl, respectively. However, the fit peaks at approximately 201.45 eV and 202.35 eV corresponded to the Cl(2p^1/2^) and Cl(2p^3/2^) of C_6_H_5_Cl, respectively, which indicated that O=C–N–Cl have been formed by N-chlorination of the amidic nitrogen, and that C_6_H_5_Cl formed due to the Orton rearrangement of O=C–N–Cl during the chlorine treatment [28,29].

### 3.4. Surface Wettability of MAF and MAFC

Figure 5 shows the water, ethylene glycol, and PU prepolymer contact angle of the MAF and MAFC. It is well known that the surface treatment of the filler plays an important role in the formation of an excellent interface with the matrix. To study the changes of the surface energies of the MAF and MAFC caused by the surface treatment, the surface wettability and surface energy of the MAF and MAFC were analyzed by contact angle measurements. The contact angles of the MAF were found to be 94.6° and 72.5° with deionized water and ethylene glycol, respectively. The water and ethylene glycol contact angles of the MAFC were found at 51.2° and 35.7°, respectively, indicating that the surface wettability of the MAFC was improved after the chlorination treatment.

The change of the surface energies of the MAF and MAFC, as summarized in Table 3, exhibited similar trends to those of the contact angle. The 21.9 mJ/m^2^ total surface energy ($\gamma_{S}$) of the MAF was composed of a 3.6 mJ/m^2^ polar component ($\gamma_{S}^{p}$) and a 18.3 mJ/m^2^ dispersive component ($\gamma_{S}^{d}$). However, the total surface energy (*γ**_S_*) of the MAFC increased to 49.1 mJ/m^2^, which consisted of a 40.3 mJ/m^2^ polar component ($\gamma_{S}^{p}$) and a 8.8 mJ/m^2^ dispersive component ($\gamma_{S}^{d}$) after the chlorination treatment. The surface polar component ($\gamma_{S}^{p}$) of the MAFC has exhibited a drastic improvement from 3.6 to 40.3 mJ/m^2^; this improvement was attributed to the presence of polar functional groups, such as –Cl–, –COOH, and –NH_2_, which were incorporated on the surfaces of the MAFC after chlorination treatment. In addition, the PU prepolymer contact angles before chlorination and after modified chlorination decreased from 70.9° to 30.7°, as shown in Figure 5e,f, illustrating that the wettability and interfacial bonding of the fibrids and PU matrix improved after chlorination modification.

### 3.5. Mechanical Properties of PU Elastomer Composites 

Figure 6 shows the mechanical properties of the composites: (a) tensile strength, (b) tear strength of MAF/PU and MAFC/PU, (c) tensile stress-strain curve of MAF/PU, and (d) tensile stress-strain curve of MAFC/PU. As shown in Figure 6a,b, the tensile and tear strength of the MAF/PU and MAFC/PU composites increased gradually with the increasing of the filler content because of the strengthening effect of the filler. However, the larger filler content resulted in a greater number of defects in the polyurethane composites, which led to stress concentration and the occurrence of microcracks. Consequently, the tensile and tear strengths declined. Figure 6a,b shows that the tensile and tear strengths of the MAF/PU composite with 1 wt% MAF reached the maximum values of 30.6 MPa and 70.0 kN·m^−1^, respectively. In contrast, the MAFC/PU composite had higher tensile and tear strengths with 1 wt% MAFC content. Furthermore, the tensile and tear strengths of the MAFC/PU composite with 1.5 wt% MAFC reached maximum values of 36.4 MPa and 80.1 kN·m^−1^, respectively, corresponding to improvements of 43.3% and 21.1%. The tensile results indicated that the mechanical properties of the MAFC/PU performed superior to those of the MAF/PU at the same fibrid weight fraction, indicating that the interfacial adhesion between the MAFC and PU matrix was improved by introducing polar groups due to the MAFC surface chlorination treatment. The tensile stress-strain curves of the MAF/PU and MAFC/PU composites are illustrated in Figure 6c,d, respectively. The MAF/PU and MAFC/PU composites exhibited similar stress-strain behaviors. In the initial stage, the stresses in the PU elastomer composites increased linearly at low strain levels and exhibited a certain rigidity; the stresses in the PU elastomer subsequently began to increase slowly with strain; in the final stage, the stress increased rapidly due to the strain-hardening effect until the polyurethane composites fractured [30]. The stress-strain curve of the MAF/PU composite changed little with the addition of the MAF, whereas the stress-strain curve of the MAFC/PU composite changed significantly with the addition of the MAFC, which indicated that the modified MAFC produced a more pronounced strengthening and toughening effect on the polyurethane elastomers.

Table 4 shows the mechanical properties of the MAF/PU and MAFC/PU composites. The composites were determined by tensile tests. The hardness values, tensile strengths, tear strengths, elongations at the break, 100% moduli, and 300% moduli of PU-MAF and PU-MAFC composites were shown in Table 4. The tensile and tear strength of the MAF/PU and MAFC/PU composites were enhanced remarkably. To study the mechanical properties, five specimens were chosen from each of the eight samples.

### 3.6. Morphologies of MAFC/PU Composites

To observe the distribution of the MAFC in the MAFC/PU composites, the ends of the MAFC/PU composite samples were immersed in a 50% sulfuric acid solution for 30 min, and the surface polyurethane was etched to expose the internal MAFC. Figure 7 shows the SEM images of the surface etching morphology of the MAFC/PU composites. As shown in Figure 7a, the distribution of MAFC in the PU-FC-1.5 was visible after etching the polyurethane (PU). The MAFC was distributed as a disordered manner in the MAFC/PU composites, and no physical cross-linking was found between the MAFC. As shown in Figure 7b, the surface of the MAFC of the MAFC/PU composite was covered with a layer of polyurethane, which indicated that the chlorinated MAFC has better interfacial adhesion with the polyurethane matrix due to the improvement of wettability after the chlorination modification.

## 4. Conclusions

It can be concluded that a novel chlorination treatment of MAF was investigated in this study. By employing this methodology, certain chlorine-containing chemical groups, O=C–N–Cl and C_6_H_5_Cl, were formed on the surfaces of the MAF due to the N-chlorination reaction of amide nitrogen and the Orton rearrangement of O=C–N–Cl, respectively, which changed the polarities of the fibers significantly. The surface wettability and surface free energy of the MAF was improved extensively, which was beneficial for forming a stronger chemical bond between the MAF and polyurethane matrix. Therefore, the mechanical properties of the polyurethane reinforced with the chlorination modified MAF (MAFC) were better than those of the MAF/PU composites. Thus, this is a novel method for obtaining high-performance polyurethane.

## Figures and Tables

**Figure 1 polymers-11-01794-f001:**
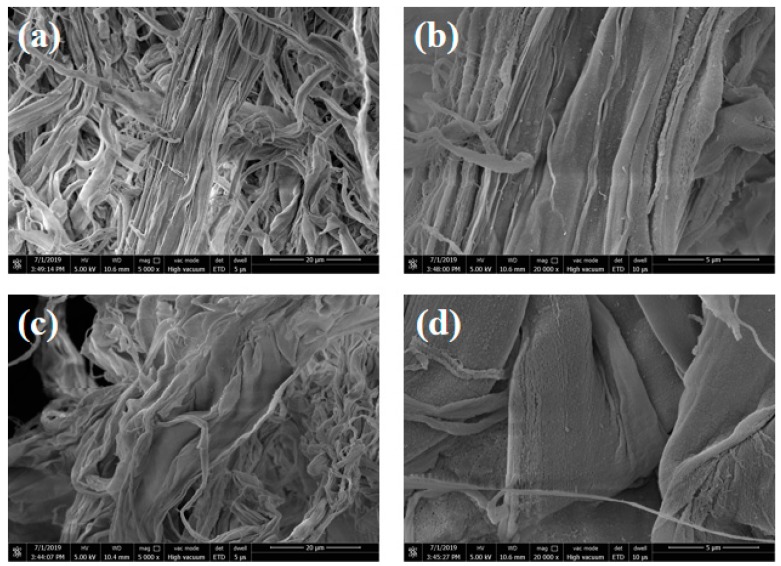
SEM images of original and modified fibrids; (**a**,**b**) meta-aramid fibrids (MAF), and (**c**,**d**) chlorinated MAF (MAFC).

**Figure 2 polymers-11-01794-f002:**
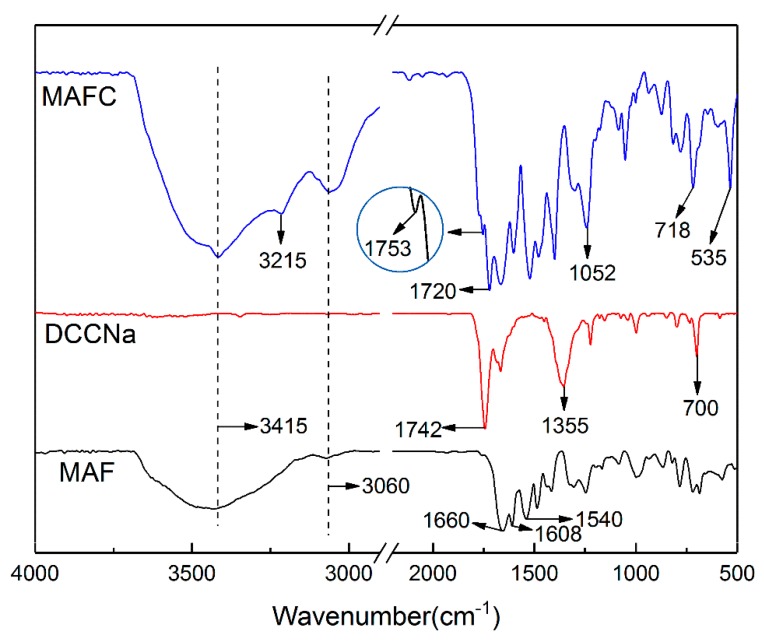
Fourier-transform infrared spectroscopy (FTIR) spectra of MAF, sodium dichloroisocyanurate solution (DCCNa), and MAFC.

**Figure 3 polymers-11-01794-f003:**
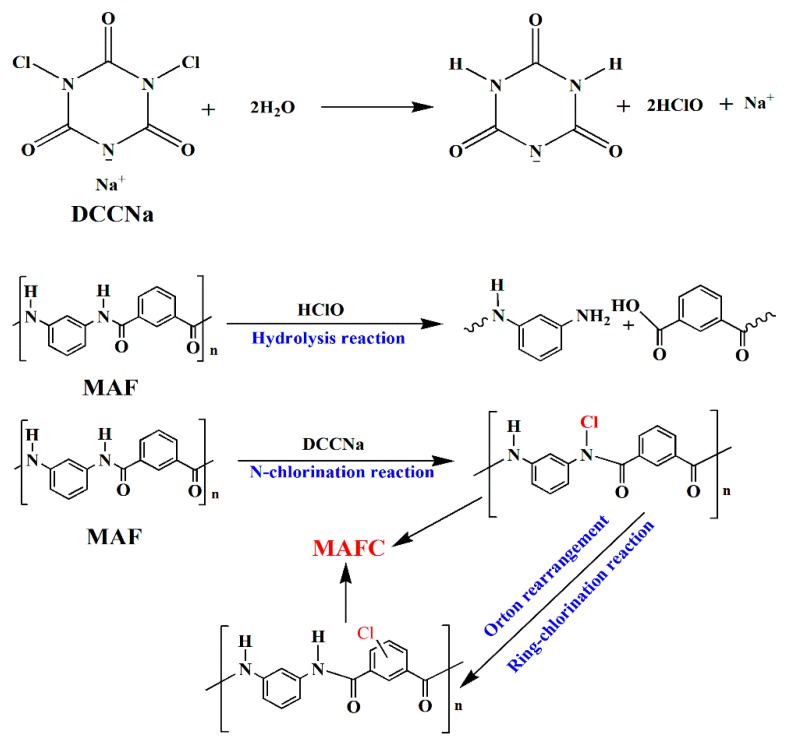
Mechanism of chlorination treatment in MAFC.

**Figure 4 polymers-11-01794-f004:**
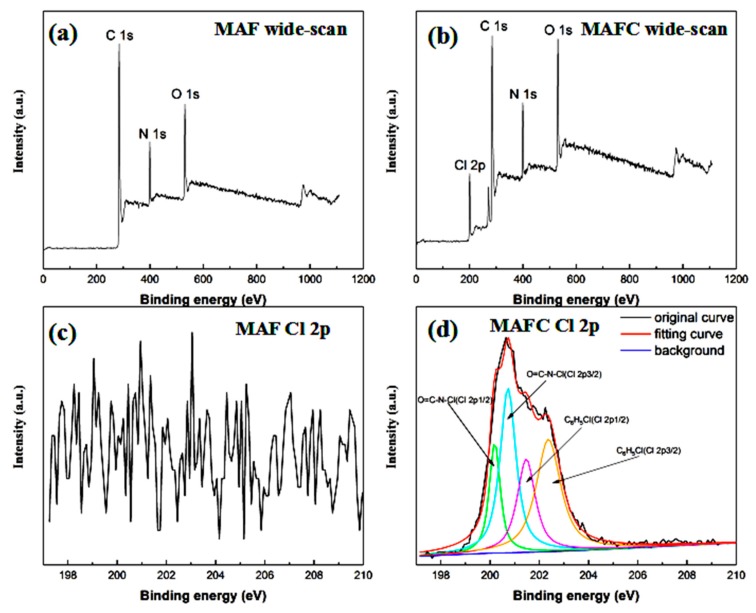
X-ray photoelectron spectroscopy (XPS) wide-scan spectra and Cl 2p spectra of (**a**,**c**) MAF and (**b**,**d**) MAFC.

**Figure 5 polymers-11-01794-f005:**
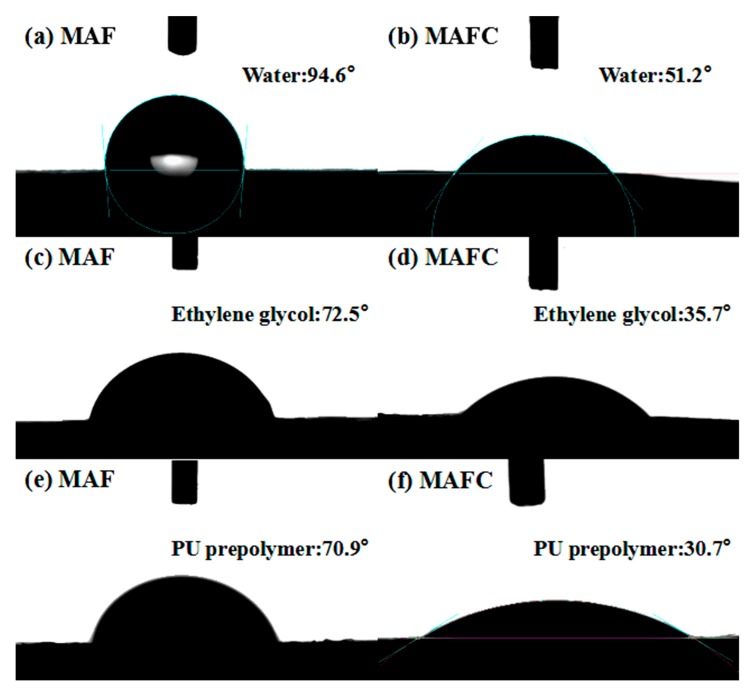
Water, ethylene glycol, and PU prepolymer contact angles of (**a**,**c**,**e**) MAF and (**b**,**d**,**f**) MAFC.

**Figure 6 polymers-11-01794-f006:**
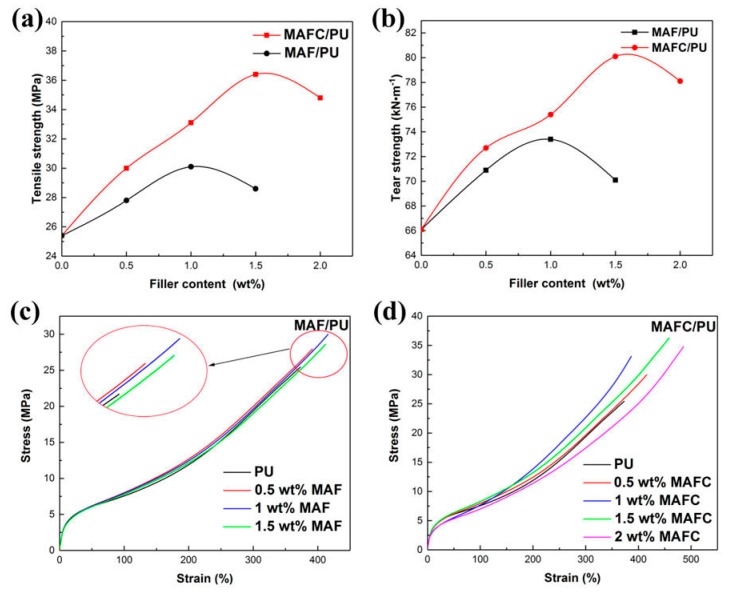
Mechanical properties of the composites: (**a**) tensile strength, (**b**) tear strength of MAF/polyurethane (PU) and MAFC/PU, (**c**) tensile stress-strain curve of MAF/PU, and (**d**) tensile stress-strain curve of MAFC/PU.

**Figure 7 polymers-11-01794-f007:**
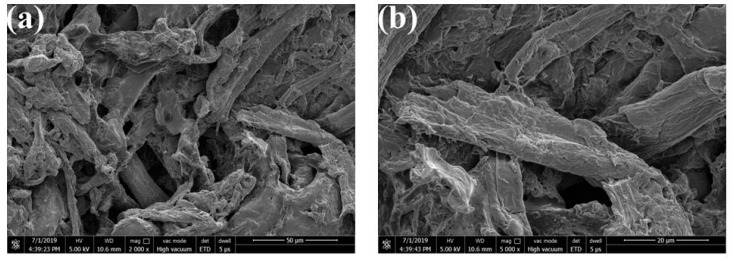
SEM images of surfaces etching of MAFC/PU in 1.5 wt% of the MAFC weight content (PU-FC-1.5) with (**a**) magnification at 2000, and (**b**) magnification at 5000.

**Table 1 polymers-11-01794-t001:** Formulations of polyurethane (PU) composites.

Sample Code	MAF/wt%	MAFC/wt%
PU	-	-
PU-F-0.5	0.5	-
PU-F-1	1	-
PU-F-1.5	1.5	-
PU-FC-0.5	-	0.5
PU-FC-1	-	1
PU-FC-1.5	-	1.5
PU-FC-2	-	2

F and FC refer to MAF and MAFC respectively.

**Table 2 polymers-11-01794-t002:** XPS surface element analysis of MAF before and after treatment.

Samples	Chemical Composition [%]				Atomic Ratio	
	C	N	O	Cl	O/C	N/C
MAF	80.9	9.5	9.6	0	0.119	0.118
MAFC	70.2	9.2	12.7	7.9	0.181	0.131

**Table 3 polymers-11-01794-t003:** Contact angles and surface free energy of MAF and MAFC.

Samples	Contact Angle *θ* ± SD^#^(°)			Surface Free Energy (mJ/m^2^)		
	Water	Ethylene Glycol		$\gamma_{S}^{d}$	$\gamma_{S}^{p}$	$\gamma_{S}$ = $\gamma_{S}^{d}$ + $\gamma_{S}^{p}$
MAF	94.6 (1.4)	72.5 (1.2)		18.3	3.6	21.9
MAFC	51.2 (2.0)	35.7 (2.0)		8.8	40.3	49.1

^#^ SD, standard deviation.

**Table 4 polymers-11-01794-t004:** Variations in mechanical properties of polyurethane composites with the increasing modified meta-aramid fibrids content.

Sample	Hardness/Shore A	Modulus at 100%/MPa	Modulus at 300%/MPa	Tensile Strength/MPa	Elongation at Break/%	Tear Strength/kN·m^−1^	Rebound Rate/%
PU	86(1)	5.6(0.2)	9.0(0.4)	25.4(1.3)	373(20)	66.1(1.2)	40(1)
P-F-0.5	87(1)	5.7(0.1)	8.6(0.4)	27.8(0.9)	391(15)	68.9(0.8)	40(2)
P-F-1	88(2)	5.0(0.3)	8.0(0.2)	30.1(2.0)	415(16)	70.0 (1.1)	39(1)
P-F-1.5	85(1)	5.8(0.4)	8.8(0.6)	28.6(1.8)	412(21)	68.1(2.0)	38(1)
P-FC-0.5	87(1)	5.9(0.2)	9.6(0.5)	30.0(1.5)	416(19)	72.7(1.6)	39(1)
P-FC-1	88(1)	6.0(0.3)	10.1(0.4)	33.1(2.1)	387(22)	75.4(2.0)	38(1)
P-FC-1.5	89(2)	6.2(0.4)	10.4(0.2)	36.4(1.4)	458(18)	80.1(1.9)	38(2)
P-FC-2	88(1)	6.2(0.3)	10.5(0.3)	34.8(1.9)	485(20)	78.1(2.3)	37(1)

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
