# Peer review of "Chlorination Treatment of Meta-Aramid Fibrids and Its Effects on Mechanical Properties of Polytetramethylene Ether Glycol/Toluene Diisocyanate (PTMEG/TDI)-Based Polyurethane Composites"

_polymers, 2019, doi:10.3390/polym11111794_

Round 1

Reviewer 1 Report

This paper has been well prepared and can be accepted by the present form.

Author Response

Point 1: This paper has been well prepared and can be accepted by the present form.

 Response 1: Thank very much for your kind and valuable comments to publish this research paper in “polymers”.

Reviewer 2 Report

The manuscript reports on a novel method to increase the compatibility of aramid fibrids with PU matrix. As a general comment, the authors addressed the issue in detail, presenting results coming from different experimental techniques. However there are some issues that should be addressed before the paper can be accepted for publication:

English language needs to be improved; Page 2 line 47: “The researcher defined…”, who? Page 3 line 100: “…under certain conditions”. The conditions should be specified; Page 4: the authors mention an increase in surface roughness after chlorination but from SEM micrographs provided it is difficult to be detected; Page 6 line 169: “sand MAFCs”. The meaning is not clear; Page 7 lines 198 and 200: some angles are wrong. What is the standard deviation of these measurements?; Figure 6b: It should be “kN…”; How many sample were mechanically tested? In table 4 standard deviations should be included. “Shao should be Shore”, and “kN instead of KN”; The etching procedure should be described; Figure 7: Higher magnifications should be provided to appreciate differences.

Author Response

Manuscript Number: polymers-610201

“Chlorination treatment of meta-aramid fibrids and its effects on mechanical properties of polytetramethylene ether glycol/toluene diisocyanate (PTMEG/TDI)-based polyurethane composites”

 Thank you very much for your kind and valuable comments of our paper. The reviewer comments have been revised for kind looking. The revised texts have been marked with Red and Yellow colour in the manuscript. I would like to request the reviewer to have screening the manuscript. Changed and revised responses shown in below for your kind looking.

Reviewer 3 Report

Manuscript entitled "Chlorination treatment of meta-aramid fibrids and its effects on mechanical properties of polytetramethylene ether glycol/toluene diisocyanate (PTMEG/TDI)-based polyurethane composites" is well written and presents very good research and going to be interesting for the reader's. 

Author Response

Manuscript entitled "Chlorination treatment of meta-aramid fibrids and its effects on mechanical properties of polytetramethylene ether glycol/toluene diisocyanate (PTMEG/TDI)-based polyurethane composites" is well written and presents very good research and going to be interesting for the reader's. 

Response: Thank very much for your kind and valuable comments to publish this research paper in “polymers”.

Round 2

Reviewer 2 Report

The authors have revised the manuscript but there are some issues that still need to be addressed as follows:

The standard deviations should be added in table 4 to check the dispersion of data; The quality of written English needs to be significantly revised and improved.

Author Response

Thank you very much for your kind and valuable comments of our paper. The reviewer comments have been revised for kind looking. The revised texts have been marked with blue colour in the manuscript. I would like to request the reviewer to have screening the manuscript. Changed and revised responses shown in below for your kind looking.

if we need any correction again pls be advised us. Thanks
